# A New Full Digital Workflow for Fixed Prosthetic Rehabilitation of Full-Arch Edentulism Using the All-on-4 Concept

**DOI:** 10.3390/medicina60050720

**Published:** 2024-04-26

**Authors:** João Martins, João Rangel, Miguel de Araújo Nobre, Ana Ferro, Mariana Nunes, Ricardo Almeida, Carlos Moura Guedes

**Affiliations:** 1Prosthodontic Department, Malo Clinic, Avenida dos Combatentes, 43, Level 10, 1600-042 Lisbon, Portugal; jmartins@maloclinics.com (J.M.); jrangel@maloclinics.com (J.R.); ralmeida@maloclinics.com (R.A.); cguedes@maloclinics.com (C.M.G.); 2Research, Development and Education Department, Malo Clinic, Avenida dos Combatentes, 43, Level 11, 1600-042 Lisbon, Portugal; 3Oral Surgery Department, Malo Clinic, Avenida dos Combatentes, 43, Level 9, 1600-042 Lisbon, Portugal; aferro@maloclinics.com (A.F.); mnunes@maloclinics.com (M.N.)

**Keywords:** edentulous jaw, maxilla, dental implants, immediate loading, dental prosthesis design, diagnostic techniques and procedures, CAD-CAM

## Abstract

(1) *Background*: Recent digital workflows are being developed for full-arch rehabilitations supported by implants with immediate function. The purpose of this case series is to describe a new digital workflow for the All-on-4 concept. (2) *Methods*: The patients were rehabilitated using the All-on-4 concept with a digital workflow including computerized tomography scanning, intra-oral scanning, and CAD-CAM production of the temporary prosthesis, with the 3D printing of stackable guides (base guide, implant guide, and prosthetic guide). The passive fit of the prostheses and the time to perform the rehabilitations were evaluated. (3) *Results*: The digital workflow allowed for predictable bone reduction, the insertion of implants with immediate function, and the connection of an implant-supported prosthesis with immediate loading. The time registered to perform the full-arch rehabilitations (implant insertion, abutment connection, prosthesis connection) was below 2 hours and 30 min. No passive fit issues were noted. (4) *Conclusions*: within the limitation of this case series, the digital workflow applied to the All-on-4 concept using stackable base-, implant-, and prosthetic guides constitutes a potential alternative with decreased time for the procedure without prejudice of the outcome.

## 1. Introduction

The All-on-4 concept, a full-arch rehabilitation approach using implants with immediate function (2 anterior implants inserted in an axial position and 2 posterior implants inserted with distal tilting), is a treatment alternative validated for the short-, middle- and long-term outcomes [1,2] and using different implant micro-designs [1,2,3] and prosthodontic materials [4,5,6], with cumulative implant survival rates ranging between 93% and 100% (1 to 18 years of follow-up) and marginal bone losses between 0.28 mm and 2.23 mm (1 to 15 years of follow-up) [1,2,3].

In the current millennium, there has been a significant development in implant dentistry concerning the use of digital tools. Digital tools such as guided surgery and recently navigated surgery provide a potential significant gain in the predictability of the procedures and patient safety, allowing good treatment outcomes in short- and mid-term follow-ups with 94.5% to 100% cumulative implant survival rates [7,8] and 0.53 mm of marginal bone loss [7]. A recent development in implant dentistry represents the use of fully digital workflows, comprising three-dimensional imaging, computer-guided implant insertion, digital impressions using intra-oral scanners, and CAD-CAM prostheses [9,10,11]. This development has been challenging over the years given the potential issues in the acquisition of data when using intra-oral scanners and the registration and superimposition of data in the absence of natural teeth or other anatomical references [12]. These potential issues have deserved some attention from the scientific/clinical community with the improvement in the workflow by introducing stackable guides that allow the connection of surgical guides and an interim poly(methylmethacrylate) (PMMA)-fixed prosthesis following CAD-CAM digital design and 3D printing [13,14,15,16,17,18]. Nevertheless, several limitations arise from this workflow, concerning angular deviation differences up to almost 10 degrees between planning and post-operative implant positioning [14,17]; the stability of markers and patient comfort when using the patient’s face for anchoring [15]; or an increased treatment cost and CAD knowledge [18]. Fully digital workflows offer potential decreased rehabilitation times and increased patient satisfaction given their less invasive nature [16,19], with a highlight on the significant percentage of patients (91%) preferring flapless implant placement [19]. Nevertheless, more studies are necessary for full scientific validation.

The aim of this clinical report was to describe a digital workflow for immediate function full-arch rehabilitations through the All-on-4 concept.

## 2. Materials and Methods

The present case series illustrates the use of stackable 3D-printed guides in the surgical and prosthetic workflows for standard All-on-4 concept rehabilitations (Nobel Biocare AB, Gothenburg, Sweden). In accordance with the Declaration of Helsinki, the patients provided written informed consent for participating and for the publication of this report. This case series received approval from an independent ethical committee (Ethical Committee for Health, Lisbon, Portugal; authorization 003/2023).

Four patients presenting esthetic and functional concerns were referred to the clinic for full-arch rehabilitation: Patient 1, a 55-year-old female patient in need of a maxillary rehabilitation; patient 2, a 28-year-old female patient in need of a mandibular full rehabilitation; patient 3, an 81-year-old male patient in need of a mandibular rehabilitation; and patient 4, a 52-year-old female patient in need of a maxillary rehabilitation. The patients’ medical charts were analyzed. Patient 1 had a history of cancer and smoked 5 cigarettes per day, and patient 4 smoked 10 cigarettes per day. The remaining patients were healthy.

### Description of the Technique

An orthopantomography and a computerized tomography (CT) scan were performed for the patients’ baseline data collection. In addition, an intra-oral scanning (Trios, 3 shape A/S, Copenhagen, Denmark) was also performed (Figure 1 of patient 1).

Using the collected information, the interim prosthesis and the guides were digitally designed using the following software: Exocad (DentalCAD 3.1 Rijeka, Exocad GmbH, Darmstadt, Germany) software was used for the digital prosthetic design; BlueSky Plan (V4.9.4 64 bit, BlueSkyBio, Libertyville, IL, USA) was used for digital bone reduction and dental implant planning; and Meshmixer (V3.1, Autodesk, San Francisco, CA, USA) was used to design the base guide, surgical guide, and prosthetic guide. The components were then printed using a 3D printer (Formlabs Inc., Form 3B+, Somerville, MA, USA). 

After printing the interim prosthesis, a silicone matrix of the prosthesis was made using Furbo Alu Flask, Vertys Orange Plus, Vertys High Security Fluid, and Vertys Easy Putty silicones (Vertysystem, Altavilla Vicentina, Italy). Following the creation of the silicone matrix, Vertys Prothesis Excel—Orange pink and Vertys Templus MM2 Monomassa (Vertysystem) were injected into the flask aiming to increase both resistance and esthetic levels. Considering the reverse planning, the first item to be designed was the prosthesis which would then indicate the amount of bone reduction needed and the implants’ position. Bone reduction was planned on BlueSkyBio, (Libertyville, IL, USA) prior to the implant planning, taking into consideration the crest shape as well as the patient’s smile line and prosthetic space. Following this procedure, the Base Guide was designed using BlueSky Plan (BlueSkyBio) and Meshmixer (Autodesk). The implant guide was designed on Meshmixer (Autodesk), after planning the implants on BlueSky Plan (BlueSkyBio) according to the patient’s available bone and maxillary anatomy. The last guide to be designed on Meshmixer (Autodesk) was the prosthetic guide that would fit the base of the temporary prosthesis and allow it to be relined in the planned position. All guides were then 3D printed (Formlabs Inc.) using BioMed Clear V1 resin (Formlabs Inc.) with 3 mm of layer thickness. Figure 2 (patient 1) illustrates the planning step previously described.

Magnets and a male–female plug connection were used to link the guides (male plug) to the base guide (female plug). This improved stability and led to an easier positioning [18] and accuracy between guides.

Considering the surgical protocol, the intervention was performed using local anesthesia with articaine chlorhydrate with epinephrine 1:100,000 (Scandinibsa 2%, Inibsa Laboratory, Barcelona, Spain). Prior to surgery, the patients were administered diazepam (Valium 10 mg, Roche, Amadora, Portugal). Antibiotic medications (amoxicillin 875 mg + clavulanic acid 125 mg, Labesfal, Campo de Besteiros, Portugal) were given 1 h before surgery and daily for 6 days thereafter. Corticosteroids (prednisone 5 mg, Meticorten Schering-Plough Farma Lda, Agualva-Cacém, Portugal) were administered daily in a regression mode (15 to 5 mg) between the day of surgery and 4 days post-operatively. Anti-inflammatories (ibuprofen, 600 mg, Ratiopharm Lda, Carnaxide, Portugal) were given between day 4 and 7 post-operatively. Analgesic medication (clonixine 300 mg, Clonix, Janssen-Cilag Farmaceutica Lda, Barcarena, Portugal) was given on the day of surgery and only if needed on the first 3 days post-operatively. Antacids (Omeprazole, 20 mg, Lisbon, Portugal) were administered between the day of surgery and day 6 post-operatively. A mucoperiosteal flap was raised along the top of the ridge with relieving incisions on the buccal aspect in the molar area.

The flap was raised to allow the base guide to be inserted and perfectly fitted to the buccal cortical. This guide was fixed to the bone using 2.2 mm drills and guide fixation pins (BlueSkyBio, Libertyville, IL, USA). This guide was kept fixed until the end of the treatment since it steered bone reduction and positioned the remaining guides. The next step consisted of performing bone regularization/reduction using a Piezomed SA-320 unit (W&H, Burmoos, Austria), a Rounger Beyer (Hu-Friedy, Chicago, IL, USA), and a round handpiece bur (Axis Dental, Crissier, Switzerland) in order to obtain a stable and flat bone surface. Once bone regularization was completed, the implant guide was stacked over the base guide, allowing the guided placement of the implants in accordance with the planning in BlueSkyBio software (V4.9.4 64 bit, BlueSkyBio, Libertyville, IL, USA). Two NobelSpeedy Groovy regular platform implants (4 mm in width and 10 mm in length) (Nobel Biocare AB, Gothenburg, Sweden) and two NobelSpeedy Groovy regular platform implants (Nobel Biocare AB) of 13 mm in length were used for the anterior straight implants and the posterior angulated implants, respectively. All implants had insertion torques greater than 35 N/cm. Multi-unit plus abutments (Nobel Biocare AB) were attached to the implants under the following conditions: 30° 4 mm of height on the posterior implants and straight 2 mm abutments on the anterior implants, using the All-on-4 concept (Nobel Biocare AB).

After implant insertion, the implant guide was removed, and the prosthetic guide was stacked on top of the base guide. This guide allowed for relining the temporary prosthesis in the previously planned position. The surgeon removed the base guide and sutured the flap using non-resorbable sutures (3-0, B Braun Silkam, Aesculap Inc., Center Valley, PA, USA). The prosthesis was relined in the mouth using a pink acrylic-resin (Unifast, GC Corporation, Tokyo, Japan). Following the relining procedure, the prosthesis was finished chairside without laboratorial support. According to patient desires, the definitive prosthodontic work consisted of a metal–ceramic prosthesis [titanium framework and all-ceramic crowns [NobelProcera titanium framework, NobelProcera crowns (Nobel Rondo ceramics; Nobel Biocare AB)] or metal–acrylic prosthesis [titanium framework (NobelProcera titanium framework; Nobel Biocare AB) and acrylic resin prosthetic teeth (Mondial and Premium teeth, Heraeus Kulzer GmbH, Hanau, Germany)]. Figure 3 (patient 1) describes the surgical step. Figure 4 (patient 1) describes the post-surgical assessment. Figure 5 illustrates a control periapical radiograph after the rehabilitation process. Figure 6 illustrates the main steps of rehabilitation for patient 2 (mandibular full-arch rehabilitation).

Outcome measures were evaluated at implant insertion/prosthesis connection on the day of surgery and at 10 days, 2 months, 4 months, and 6 months post-surgically. Primary outcome measures were implant and prosthetic survival based on function: implants removed and prostheses needed to be replaced were considered failures. Secondary outcome measures were evaluated at different time points. The time elapsed to perform the rehabilitation was evaluated on the day of surgery and measured from the beginning of surgery to the connection of the prosthesis. The passive misfit of the prosthesis (registered as present or absent) was evaluated through the one-screw test at 10 days post surgery. Radiographic assessment through periapical radiographs was performed at 10 days and 4 months post surgery. The incidence of mechanical complications (fracture or loosening of any prosthetic component) and biological complications (infection, abscess, or fistulae) was evaluated at 10 days and 2, 4, and 6 months post surgery. 

## 3. Results

All implants and prostheses remained in function throughout the functional osseointegration period. No passive misfit issues were noted on the prostheses upon radiographic evaluation. The total time elapsed to perform the rehabilitation was on average 2 h and 30 min (range: 2 h and 20 min to 2 h and 45 min). Upon radiographic evaluation, no issues were noted at 4 months. Patient 1 presented an abscess on the second quadrant during the first month of follow-up that did not involve the implants. The patient admitted having smoked during the post-operative period. The abscess was treated through the administration of antibiotics (amoxicillin 875 mg + clavulanic acid 125 mg, Labesfal; every 12 h for a period of 8 days), with confirmed uneventful healing. No other biological complications occurred during the follow-up of this study for the remaining patients. There was an absence of mechanical complications throughout the evaluation period.

## 4. Discussion

The digital workflow described in this case series allowed the correct placement of the implants with good primary stability and the accurate connection of an implant-supported fixed prosthesis while significantly reducing the time required for the rehabilitation procedure. Furthermore, the goal was achieved without compromising the predictability, considering the standards of a full-arch rehabilitation using the All-on-4 concept (Nobel Biocare AB), allowing immediate function through an interim prosthesis [20].

The full-arch rehabilitation of the edentulous maxilla through an implant-supported fixed prosthesis comprises several alternatives to the All-on-4 concept (Nobel Biocare AB), through the potential use of standard implants anchored in the maxillary tuberosity or pterygoid region. A previous systematic review on the placement of dental implants in the maxillary tuberosity estimated a 94.6% implant cumulative survival rate with a follow-up up to 14 years [21]. A recent systematic review assessing the outcome of implants in the pterygoid region estimated a 95.5% implant cumulative survival rate at 6 years of follow-up [22]. As illustrated by the results, both these treatment modalities are viable with predictable outcomes, offering the surgical advantage of a virtual impossibility of implant insertion in the posterior region of the maxilla. Compared to these treatment modalities, the All-on-4 concept (Nobel Biocare AB) is potentially the most documented treatment alternative in implant dentistry, where a previous long-term study documented a 97.6% and 94.7% implant cumulative survival rate at 6 and 14 years, respectively [1], whereas a recent systematic review estimated a 97.1% cumulative implant survival rate at 13 years specifically for tilted implants in the maxillary rehabilitations [23]; both studies evidenced higher survival rates at similar time points when compared to tuberosity or pterygoid anchored implants. Nevertheless, it is possible to apply the present digital workflow to any type of full-arch implant-supported treatment as long as there is available bone in the maxilla to support the base guide.

The potential accuracy achieved through digital planning and execution is a significant benefit, as it ensured advantages not only on the surgical aspect (bone reduction, implants placed in the optimal location for stability and long-term success), but also on the functional and esthetic aspects with the digitally planned prosthetic teeth color, shape, and occlusion matching the real prosthesis. As described in a previous meta-analysis and randomized controlled trial, freehand osteotomy and implant placement offer less predictability, making pilot drill and fully guided approaches much safer and accurate [24,25]. A systematic review and meta-analysis comparing virtual treatment planning and outcome in different setups registered a significantly lower accuracy in clinical and cadaver setups compared to in vitro studies [26]. Additionally, no significant differences in apical–coronal deviation or vertical deviation were registered [26]. 

To the authors’ knowledge, this is the first manuscript reporting on the use of three guides attached magnetically: the base guide, surgical guide, and prosthetic guide. Previous publications documented the possibility of connecting the base guide and surgical guide pre-operatively [14]. The present protocol allows for the insertion of the base guide for bone reduction, followed by the attachment of the surgical guide to the base guide for implant-guided surgery, and finally the removal of the surgical guide and connection of the prosthetic guide to the base guide for capturing the correct prosthetic placement, for a fully guided rehabilitation process. As previously reported, the use of magnets for attaching surgical guides is a facilitating factor for their positioning and increased stabilization (given the force of attraction field between magnets), which is clinically relevant for successful full-arch-guided rehabilitations [14,18,27]. Moreover, the fact that the base guide was osseo-supported confers more stability and precision when compared to muco-supported base guides. Furthermore, the flexibility of the protocol allows the guides to be inserted or removed in any phase of execution. For example, the prosthetic guide, because it was positioned on the base of the prosthesis without interference with the occlusal aspect, could be placed immediately after bone reduction and therefore confirm not only that the bone reduction was properly performed but also that the base guide was positioned following the right occlusion plan. Additionally, the implant guide allowed for guiding the implants while increasing the surgeon’s visual field, while facilitating saline irrigation during the implant insertion. The implant guide can also be removed at any point during the surgery, providing the surgeon with more freedom to adapt the protocol. However, additional studies are needed to assess the accuracy of the implant placement using the magnetic surgical guide.

The current protocol benefitted from artificial intelligence input since automatic segmentation was provided through BlueSky Plan (BlueSkyBio, Libertiville, IL, USA). The procedure had a successful outcome since no complications or mechanical failures occurred, indicating the effectiveness of the digital planning approach. This provides validation of the prototype for digital planning in terms of optimal implant positioning and immediate loading. 

The recent literature reported on the outcome of implant treatment using guided implant surgery. A critical review based on randomized controlled trials investigating the clinical applications and effectiveness of guided implant surgery registered no significant differences in implant survival rate and effectiveness when compared to conventional implant placement procedures [28]. A systematic review and meta-analysis investigating the implant failure rates and corresponding association with guided and free-hand implant placement techniques reported high survival rates for both treatment modalities, despite the 4% increase in failure rate for the free-hand alternative [29]. From the interpretation of the available literature with the higher degree of evidence, it is possible to assume the safety of the protocols for guided implant surgery, finding parallels with the results also obtained in the present case series. Nevertheless, it is important to consider other variables that might impact the outcome including the operators’ clinical experience and skill, the type of technique implemented, and the necessary learning curve. 

This procedure allowed for reducing the time needed to rehabilitate the patient. The procedure took on average two and a half hours from the beginning of the surgery to the placement of the immediate bridge instead of the usual 4 to 7 h required for conventional approach [20,30]. This reduction in time is significant, as it reduces the time a patient must spend under anesthesia and at the clinic, making it more comfortable and less demanding from a physical point of view. Another item to note relates to the advantage of using stackable guides: these allow the implant positioning to be changed perioperatively, in a similar fashion as in dynamic navigation, which is impossible with static guides [31].

Despite the advantages of the digital planning approach, there are some potential downsides to consider. One limitation is the requirement for a minimum mouth opening capability of at least 50 mm to accommodate the guides used in the procedure. Additionally, the risk of inaccuracies in the production of guides is a potential concern, as it could lead to serious perioperative and postoperative complications. It is essential to address these issues to ensure that the digital planning approach continues to be a safe and effective alternative to traditional analog approaches. Moreover, additional limitations relate to the learning curve in the adaptation of the surgical and prosthetic protocols to the digital workflow; the necessary increased knowledge in CAD; and the extra pre-operative steps for data collection to perform the planning. Given these reasons, the current technique is presented as a prototype and not a measurement of accuracy. 

Future developments in the field of digital planning and execution of dental surgical procedures are promising. The use of digital planning approaches may offer a significant advantage in terms of accuracy and efficiency, and further research could help refine the process and make it more accessible to a wider range of patients. This prototype demonstrated potential benefits of the digital planning approach to the All-on-four concept (Nobel Biocare AB), with advantages to both the clinician and the patient. Future studies should investigate the accuracy between digital planning and actual implant position in full digital workflows (including navigated surgery); the prosthetic/implant survival outcome; and potential implications for perioperative complications.

## 5. Conclusions

The present protocol describes a digital workflow for full-arch rehabilitation from planning to connection of the interim prosthesis, aiming for improved efficiency. Within the limitations of this report, the digital workflow applied to full-arch rehabilitation using the All-on-4 concept (Nobel Biocare) seems to be a viable alternative to the analogical procedure, providing a faster rehabilitation time without prejudice of the outcome. The procedure allowed for a full-arch rehabilitation with an average of 2 h and 30 min. No implications were registered on the remaining outcome measures: no issues of passive fit were registered; all implants remained in function during the follow-up of this study, rendering a 100% implant survival rate; and no mechanical complications were registered. Future studies are necessary for robust validation of this prototype.

## Figures and Tables

**Figure 1 medicina-60-00720-f001:**
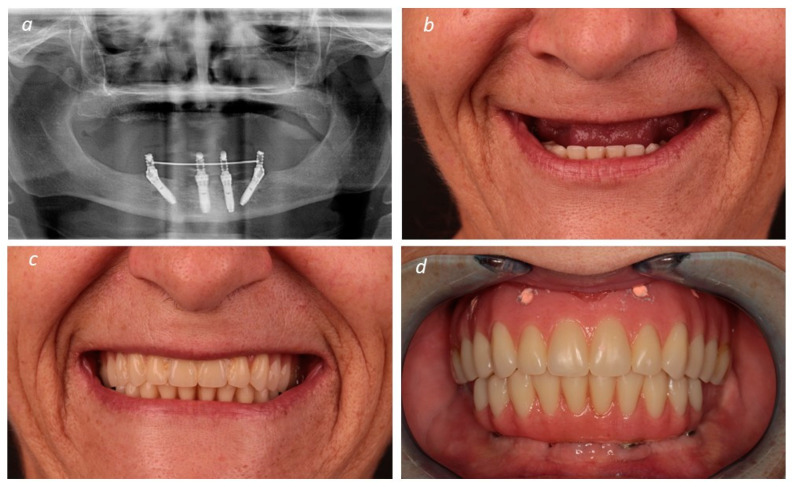
(**a**) Pre-treatment orthopantomography; (**b**) pre-treatment extraoral view at maximum smile without the prostheses; (**c**) pre-treatment extraoral view at maximum smile with the prostheses; (**d**) pre-treatment intraoral view with prostheses during occlusion.

**Figure 2 medicina-60-00720-f002:**
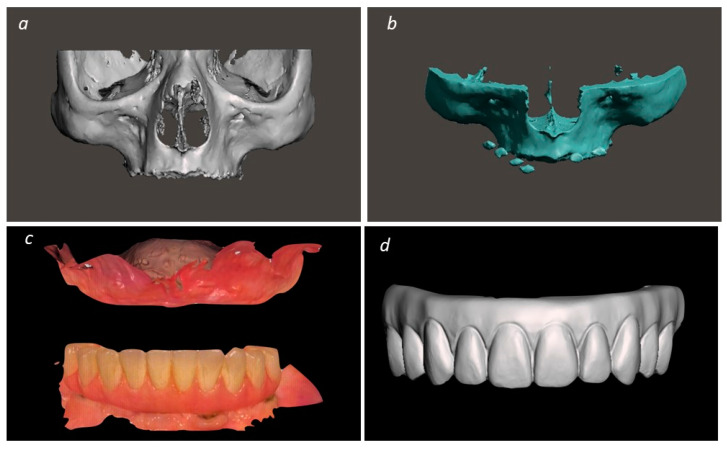
(**a**) STL file of the data collected through the CT scan; (**b**) STL file incorporating the CT scan data and the prosthesis; (**c**) intra-oral scanning image; (**d**) STL file with the designed prosthesis; (**e**) base guide design with corresponding 3D printing; (**f**) implant guide design with corresponding 3D printing; (**g**) prosthetic guide design and corresponding 3D printing; (**h**) 3D printing and finished interim prosthesis.

**Figure 3 medicina-60-00720-f003:**
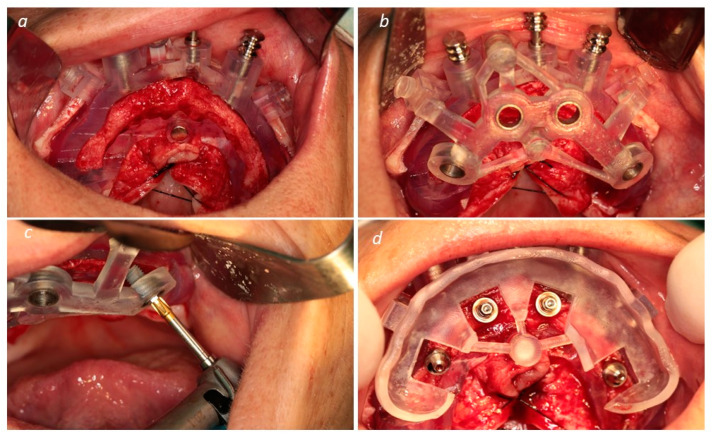
(**a**) Base guide in place; (**b**) base guide and implant guide in place; (**c**) implant-guided insertion (posterior implant tilted distally); (**d**) prosthetic guide in place after implant insertion; (**e**) prosthetic guide and interim prosthesis in place; (**f**) finished relined prosthesis.

**Figure 4 medicina-60-00720-f004:**
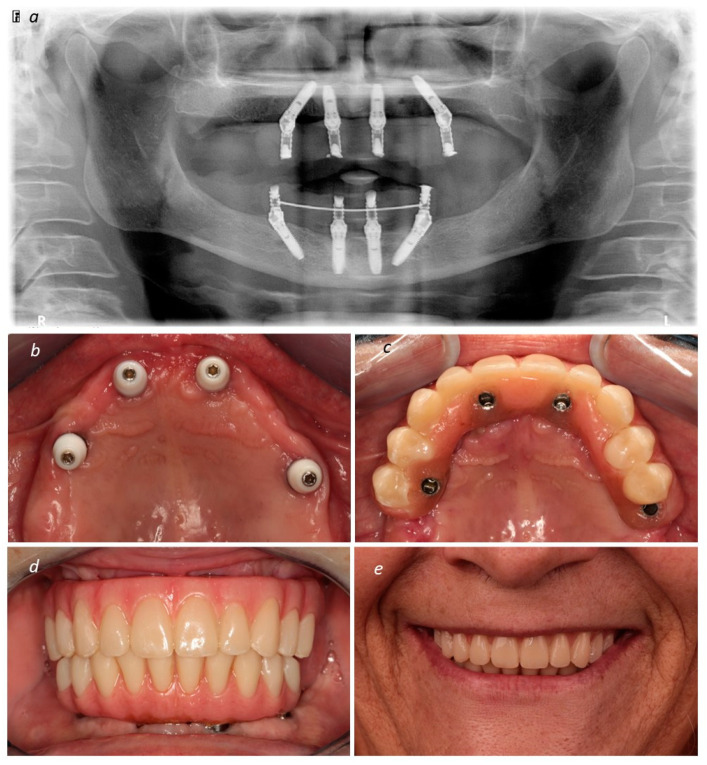
(**a**) Post-surgical orthopantomography; (**b**) intra-oral occlusal view evidencing soft tissue healing; (**c**) intra-oral occlusal view of the full-arch maxillary rehabilitation using All-on-4 concept (Nobel Biocare) evidencing the interim prosthesis; (**d**) intra-oral occlusal frontal view of the full-arch maxillary rehabilitation using All-on-4 concept (Nobel Biocare AB) with the interim prosthesis in place; (**e**) patient smiling after completion of the full-arch maxillary rehabilitation using All-on-4 concept (Nobel Biocare AB).

**Figure 5 medicina-60-00720-f005:**
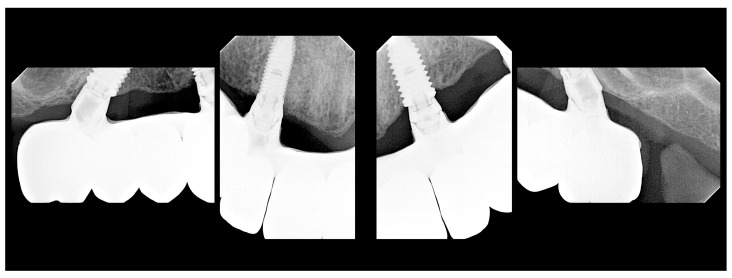
Periapical radiographs illustrating a full-arch maxillary rehabilitation using All-on-4 concept (Nobel Biocare AB) at the end of the rehabilitation process with a metal–ceramic prosthesis [titanium framework and all-ceramic crowns (NobelProcera titanium framework, NobelProcera crowns (Nobel Rondo ceramics; Nobel Biocare AB)].

**Figure 6 medicina-60-00720-f006:**
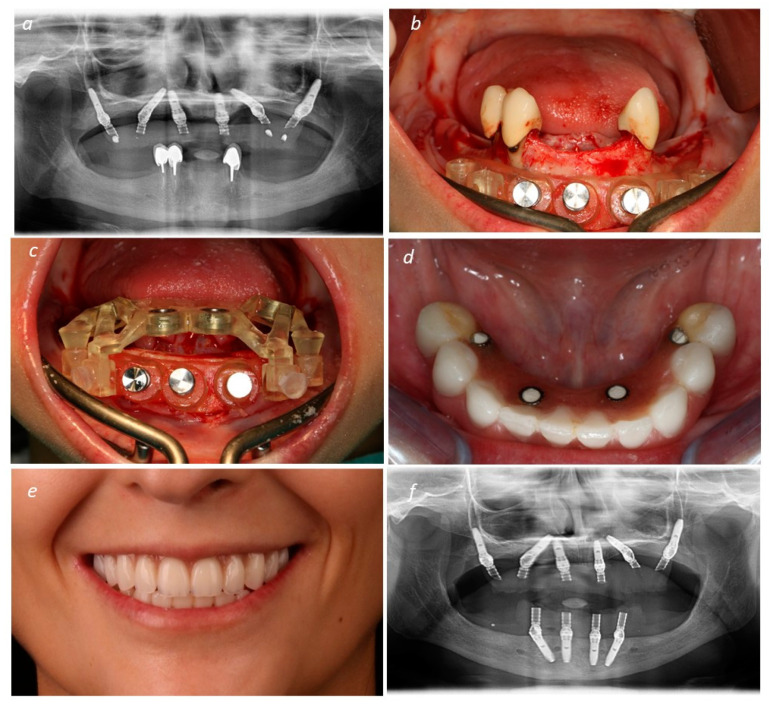
Illustration of a full-arch mandibular rehabilitation using All-on-4 concept (Nobel Biocare AB): (**a**) Pre-operative orthopantomography; (**b**) base guide in place; (**c**) base guide and implant guide in place; (**d**) intra-oral occlusal view of the full-arch maxillary rehabilitation using All-on-4 concept (Nobel Biocare) evidencing the interim prosthesis; (**e**) patient smiling after completion of the full-arch mandibular rehabilitation using All-on-4 concept (Nobel Biocare AB); (**f**) post-surgical orthopantomography.

## Data Availability

Data are available from the authors at reasonable request.

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
