# Peer review of "A New Full Digital Workflow for Fixed Prosthetic Rehabilitation of Full-Arch Edentulism Using the All-on-4 Concept"

_medicina, 2024, doi:10.3390/medicina60050720_

Round 1

Reviewer 1 Report

Comments and Suggestions for Authors

Dear authors,

The manuscript titled: „A new full digital workflow for fixed prosthetic rehabilitation  of full-arch edentulism ad modum All-on-4 Concept“ is interesting manuscript with main aim to describe a new digital workflow for the All-on-4 concept.

However I have some questions/suggestions to clarify. Please look at my notes and revise the manuscript accordingly.

In keywords the term all on 4 is mentioned twice, but all stated terms can not be found in MeSH database. Please correct this.

Introduction part is too short and I suggest to expand it. You should be focused more on scientific results from the literature regarding this topic. All what is written is too obvious and it can be read on any website giving some informations to the patient.

In section methods and materials You stated that three different softwares Exocad (DentalCAD 3.1 Rijeka, Exocad GmbH, Darmstadt, Germany), BlueSky Plan 81 (V4.9.4 64 bit, BlueSkyBio, Libertyville, IL, USA) and Meshmixer (V3.1, Autodesk, San 82 Francisco, CA, USA) were used for each phase. Can You explain why?

Can You provide clinical pictures of better quality? Those that were provided are not so clear.

What was the definitive prosthodontic work in these cases? Do you have control ortopanthomograms at the end of the rehabilitation process?

In discussion part You stated that there are no articles describing use of three guides attached magnetically, but I suggest to expand these part in more details so readers can better understand the complexity of this procedure.

Conclusion part needs revision, it is too general. Regarding the facts that were stated in this manuscript I suggest to rewrite it.

Author Response

Dear authors,

The manuscript titled: „A new full digital workflow for fixed prosthetic rehabilitation of full-arch edentulism ad modum All-on-4 Concept“ is interesting manuscript with main aim to describe a new digital workflow for the All-on-4 concept.

However I have some questions/suggestions to clarify. Please look at my notes and revise the manuscript accordingly.

  1. In keywords the term all on 4 is mentioned twice, but all stated terms can not be found in MeSH database. Please correct this.

Response: The authors thank the Reviewer’s indication. The terms were corrected as requested.

Changes: key words, lines 29 and 30.

  1. Introduction part is too short and I suggest to expand it. You should be focused more on scientific results from the literature regarding this topic. All what is written is too obvious and it can be read on any website giving some informations to the patient.

Response: The authors thank the Reviewer’s indication. The Introduction was expanded as requested focusing on results from the scientific literature.

Changes: Introduction section, lines 38-40, 45,46, 55-59, 61,62.

  1. In section methods and materials You stated that three different softwares Exocad (DentalCAD 3.1 Rijeka, Exocad GmbH, Darmstadt, Germany), BlueSky Plan 81 (V4.9.4 64 bit, BlueSkyBio, Libertyville, IL, USA) and Meshmixer (V3.1, Autodesk, San 82 Francisco, CA, USA) were used for each phase. Can You explain why?

Response: The authors thank the Reviewer’s query. Exocad was used for the digital prosthetic design; BlueSky Plan was used for the digital bone reduction and implant planning; Meshmixer was used for the design of the base guide, surgical guide and prosthetic guide. The information was inserted in the manuscript for clarity.

Changes: Materials and Methods section, lines 90-96.

  1. Can You provide clinical pictures of better quality? Those that were provided are not so clear.

Response: The authors thank the Reviewer’s query. The pictures are of high quality, maybe they were compressed by the Journal to fit the file for review, because the originals are of high quality.

Changes: None.

  1. What was the definitive prosthodontic work in these cases? Do you have control ortopanthomograms at the end of the rehabilitation process?

Response: The authors thank the Reviewer’s query. The patients are now undergoing the definitive prosthetic rehabilitation that will consist of a Metal-ceramic or metal-acrylic implant-supported full-arch restorations. The patients did not complete the definitive restorative protocol yet.

Changes: None

  1. In discussion part You stated that there are no articles describing use of three guides attached magnetically, but I suggest to expand these part in more details so readers can better understand the complexity of this procedure.

Response: The authors thank the Reviewer’s indication. The discussion was expanded so to be clear.

Changes: Discussion section, lines 247-253

  1. Conclusion part needs revision, it is too general. Regarding the facts that were stated in this manuscript I suggest to rewrite it.

Response: The authors thank the Reviewer’s indication. The Conclusion was adjusted as requested.

Changes: Conclusion section, lines 316-317, 321-323.

Reviewer 2 Report

Comments and Suggestions for Authors

An interesting alternative to the all-on-four technique. However, I would always rather opt for a maxilla reconstruction as shown in Figure 5, with the use of tuberopterygoid implants, because this type of implants could be always inserted.

 It would be of scientific importance, as a proposal the the authors to compare the results of all-on-four with the complete maxillary reconstruction with tuberopterygoid implants.

Author Response

Reviewer 2

  1. An interesting alternative to the all-on-four technique. However, I would always rather opt for a maxilla reconstruction as shown in Figure 5, with the use of tuberopterygoid implants, because this type of implants could be always inserted. It would be of scientific importance, as a proposal the the authors to compare the results of all-on-four with the complete maxillary reconstruction with tuberopterygoid implants.

Response: The authors thank the Reviewer’s indication. Despite not being the focus of the manuscript, the authors provided a brief discussion between the All-on-4 and the complete maxillary reconstruction with tuberopterygoid implants for context as requested. Nevertheless, it is possible to apply this digital workflow to any type of full-arch implant supported treatment as long as there is bone available in the maxilla to support the base guide.

Change: Discussion section, lines 216-234.